# SV-RAG: LoRA-Contextualizing Adaptation of MLLMs for Long Document Understanding

**Jian Chen**[1*]**, Ruiyi Zhang**[2*]**, Yufan Zhou**[2]**, Tong Yu**[2]**, Franck Dernoncourt**[2]
**Jiuxiang Gu**[2]**, Ryan Rossi**[2]**, Changyou Chen**[1]**, Tong Sun**[2]
University at Buffalo[1], Adobe Research[2]
`ruizhang@adobe.com`

## ABSTRACT

Multimodal large language models (MLLMs) have recently shown great progress in text-rich image understanding, yet they still struggle with complex, multi-page visually-rich documents. Traditional methods using document parsers for retrieval-augmented generation suffer from performance and efficiency limitations, while directly presenting all pages to MLLMs leads to inefficiencies, especially with lengthy ones. In this work, we present a novel framework named **S**elf-**V**isual **R**etrieval-**A**ugmented **G**eneration (SV-RAG), which can broaden horizons of *any* MLLM to support long-document understanding. We demonstrate that **MLLMs themselves can be an effective multimodal retriever** to fetch relevant pages and then answer user questions based on these pages. SV-RAG is implemented with two specific MLLM adapters, one for evidence page retrieval and the other for question answering. Empirical results show state-of-the-art performance on public benchmarks, demonstrating the effectiveness of SV-RAG.

## 1 INTRODUCTION

Documents serve as a critical medium for the preservation and dissemination of information, with millions produced annually. These documents are not limited to simple text; they encompass complex layouts and a variety of modalities such as text, tables, charts, and images. Visually-rich document understanding (VDU) is thus an essential and challenging area of research. Recently, Multimodal Large Language Models (MLLMs) has emerged, showcasing remarkable abilities to process and understand documents. These models span both proprietary and open-source domains, like GPT-4o (OpenAI, 2023), Gemini-1.5 (Team et al., 2023), and Claude-3 among closed-source models, and InternLM-XC2-4KHD (Dong et al., 2024), InternVL-Chat (Chen et al., 2023b), LLaVA-NeXT (Liu et al., 2024a), Mini-CPM (Hu et al., 2024), mPLUG-DocOwl (Ye et al., 2023b), and TextMonkey (Liu et al., 2024d) in open-source space. Their performance has been particularly notable in single-page DU tasks demonstrated on datasets like DocVQA (Mathew et al., 2021), ChartQA (Masry et al., 2022) and InfoVQA (Mathew et al., 2022).

In real-world applications, they often present documents that are much longer, containing dozens or hundreds of pages(Ma et al., 2024d; Tanaka et al., 2023; Islam et al., 2023; Zhu et al., 2021). Addressing the understanding of such lengthy documents presents MLLMs with new challenges (Ma et al., 2024d). One way is to utilize a classical document parser (Rausch et al., 2021) to extract information and formulate a prompt for LLM (Wang et al., 2023; Lamott et al., 2024), which is difficult to recover the layout in prompts and suffers performance degeneration from the document parser. The other way is to exploit the long context windows of large models, allowing them to take multiple pages at once. However, most of the input pages are not relevant to user requests, and efficiency will be compromised when the document contains hundreds of pages Ma et al. (2024d); Islam et al. (2023) or there is a document collection (Tito et al., 2021).

In this work, we first retrieve evidence pages to obtain relevant information within a vast and varied landscape of content. Unlike using a classical document parser, we propose using MLLMs as the information encoder, which have shown great generalization ability as they have been trained on a huge text corpus. After obtaining the embedding of each page, we further utilize contextualized

---

*Equal contribution, work done when JC is at Adobe Research.

late interaction for relevance scoring (Khattab & Zaharia, 2020). This design shows significantly better efficiency and accuracy than using the classical document parser to extract information. Top-$k$ pages are then selected from hundreds of pages and provided to MLLMs to answer user questions on documents.

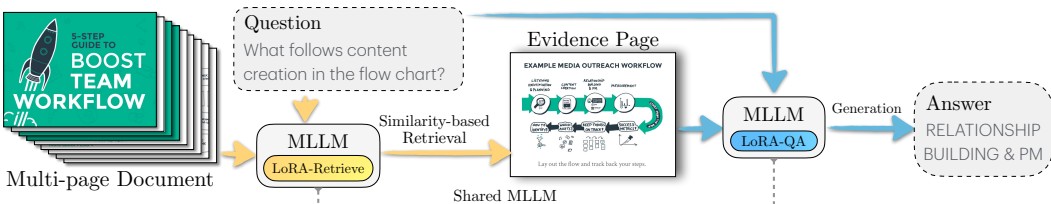

Figure 1: Overview of the SV-RAG pipeline. The multi-page document and query are encoded by a customized MLLM (yellow). The most relevant page is retrieved through similarity-based matching, and a fine-tuned MLLM (blue) generates the final answer from the evidence.

Based on this design demonstrated in Figure 1, we introduce the SV-RAG framework for multi-page document understanding, which includes modules for evidence page retrieval and answer generation. Our contributions can be summarized as follows.

- We propose a novel framework named SV-RAG to broaden the horizons of MLLMs, where we use intermediate MLLMs hidden embedding for **efficient** question-based evidence page retrieval.
- We finetune MLLMs through dual LoRA adapters for evidence page retrieval and question answering, respectively, enabling SV-RAG to be edge-friendly with great memory efficiency.
- We collect a visually-rich document QA dataset, VisR-Bench, comprising nine domains including magazine, flyer, newsletter, product manual, and presentations, etc. This dataset is built upon web-crawl documents, containing 226 documents and 471 question answer pairs.
- We empirically show that SV-RAG, with only 4B parameters, achieves state-of-the-art performance on VisR-Bench and four public benchmarks, rivaling Gemini-1.5-pro on MMLongBench-Doc and demonstrating its effectiveness.

## 2  RELATED WORK

**Visually-rich Document Understanding**   Visual Document Understanding (VDU) is the field focused on interpreting text-rich images such as tables (Zhong et al., 2019), charts (Masry et al., 2022), and webpage screenshots (Liu et al., 2024c; Tanaka et al., 2021). These images are complex, featuring a mix of text and visual elements that convey abundant information (Gu et al., 2024). To evaluate multimodal document understanding, tasks range from low-level recognition tasks, such as information extraction, to high-level cognitive tasks, such as visual question answering (Mathew et al., 2020). Models in VDU are typically divided into two categories: OCR-dependent (Xu et al., 2020) and OCR-free, based on their reliance on Optical Character Recognition (OCR). OCR-dependent models are often trained to synchronize textual and visual data. For instance, UDoP (Tang et al., 2023) is pre-trained to restore obscured textual and layout details using both image and partial text inputs. OCR-free approaches must include text recognition training. Dount (Kim et al., 2022) is an example of an OCR-free model that focuses on producing unbroken text sequences, disregarding structural details. In contrast, Pix2Struct (Lee et al., 2023a), another OCR-free model, focuses on interpreting the structure by creating HTML DOM trees from webpage screenshots. However, this technique does not easily transfer to other image types. Our method focuses on the visual question-answering task, specifically targeting questions over long documents consisting of multiple pages of multimodal information.

**Multimodal Retrieval-Augmented Generation**   Augmenting language models with information from various knowledge sources has been found to boost their performance in different NLP tasks. The Dense Passage Retriever (DPR) (Karpukhin et al., 2020) trains its retrieval mechanism with documents from within the same batch, using contrastive learning with negatively sampled examples, which enhances its capabilities in open-domain question answering. Document Screenshot Embedding (DSE) (Ma et al., 2024c) uses MLLMs as encoders for both document screenshots and queries, training through contrastive learning to achieve enhanced multimodal retrieval. Both REALM

(Guu et al., 2020) and Retrieval-Augmented Generation (RAG) (Gao et al., 2023b) consider the passages they retrieve as hidden variables and train the retrieval and generation components together, improving the efficiency of the retrieval-augmented generation approach. Taking cues from textual RAG, the Plug-and-play (Chen et al., 2024d) approach uses GradCAM (Selvaraju et al., 2020) to fetch pertinent image segments corresponding to a given query. The MuRAG (Chen et al., 2022) model introduces a multimodal retrieval-augmented Transformer that utilizes an external multimodal memory for language generation enhancement. Unlike other approaches that retrieve information from various knowledge sources, SV-RAG focuses on retrieving relevant evidence pages from a given document. This helps MLLMs generate accurate and explainable answers based on the retrieved content. MM-GEM Ma et al. (2024a) trains an MLLM with a hybrid loss for similarity computation and caption generation on natural images. In contrast, our approach targets visually rich documents, using two LoRA modules to specialize in each task.

**Multimodal Large Language Models**   While Large Language Models (LLMs) excel at text-only question answering (QA) (Dasigi et al., 2021; Lee et al., 2023b), they cannot process other modalities. To enable multimodal tasks like Visual Question Answering (VQA), MLLMs transform images and videos into visual tokens that LLMs can understand. To train these MLLMs, MiniGPT-4 (Zhu et al., 2023) leverages ChatGPT to produce data compliant with high-quality instructions, while LLaVA (Liu et al., 2023b) prompts GPT-4 with image captions and bounding boxes. Chen et al. (2023a; 2024a) have prompted OpenAI GPT-4V to generate more than 1M pieces of quality data to train MLLMs. LLaMA-Adapter (Zhang et al., 2023; Gao et al., 2023a) and mPLUG-Owl (Ye et al., 2023b) align text and image features with large-scale image-text pairs. InstructBLIP (Dai et al., 2023) has restructured 13 vision-language tasks to fit an instruction-based approach. mPLUG-Owl (Ye et al., 2023a;b) implements multi-task instruction fine-tuning with public document datasets. Recent research (Liu et al., 2023a; 2024a; Bai et al., 2023; Dong et al., 2024; Xu et al., 2024; Luo et al., 2024) improves visual encoders by increasing resolution, leading to significant advances in downstream applications but also raising memory costs, especially in multi-page tasks. TextMonkey Liu et al. (2024d) compresses visual tokens using a token resampler. Our method extends MLLMs to handle multi-page documents by retrieving relevant pages, reducing computation and distractions from long token sequences.

## 3   SV-RAG Method

Multi-page document understanding aims to answer questions related to long and complex documents containing both text and images from users. We denote a document of $n$-pages as a sequence of images, $\mathbf{X} = \{\mathbf{x}_1, \mathbf{x}_2, \ldots, \mathbf{x}_n\}$. Text token sequence of a question $q$ is denoted as $\{q_1, q_2, \ldots, q_n\}$. Traditional approaches that begin with a parsing step to extract content elements such as images, tables, and forms from documents, then generate answers based on these contents using LLMs (Saad-Falcon et al., 2023; Wang et al., 2023). Here, we first consider using MLLMs to handle this task and avoiding the heuristic document parsing process, where we directly convert each page into a single image. It is not desired, as most pages in a document are irrelevant to user questions and performing an evidence page retrieval can further enhance the efficiency.

We introduce SV-RAG, a method that efficiently leverages the capabilities of pre-trained MLLMs for long document question-answering (QA). SV-RAG can broaden the horizon of MLLMs to answer questions over long documents or document collections with hundreds of pages. This finding is based on the fact that hidden states of MLLMs can be effective page representations for question-based retrieval, as shown in Section 5.6. This representation ability can be further enhanced with contrastive training using a LoRA adapter, demonstrating surprising retrieval performance of MLLMs. Furthermore, we can finetune a LoRA-adpter of QA to further enhance the performance of SV-RAG on specific domains. In summary, we first retrieve evidence pages to rank these images based on their relevance score to a given question $q$, then select the most relevant images, which are then fed into the MLLM to generate the answer. In this section, we introduce the SV-RAG architecture in Section 3.1, retrieval training in Section 3.2 and dual-adapter designs in Section 3.3.

### 3.1   Architecture

Figure 2 presents an overview of our model architecture, which comprises two MLLM-based modules for the retrieval of evidence pages and question answers.

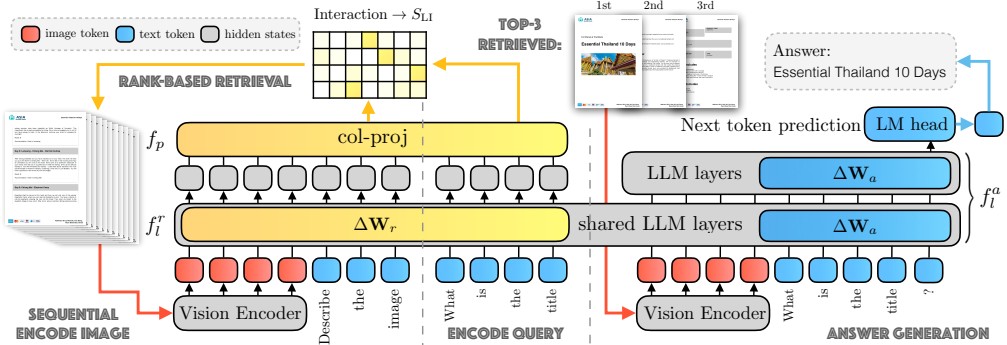

Figure 2: Model overview of SV-RAG. It contains two modules, which are finetuned using LoRA (Hu et al., 2021), sharing the **same** pretrained multimodal LLM backbone. The retrieval module selects evidence pages for the other QA module, which provides responses to user questions.

**Col-Retrieval Module**  Building on the approach introduced in ColPali (Faysse et al., 2024), we employ a modified MLLM for retrieval, comprising a vision encoder $f_v$, a large language model (LLM) $f_l^r$, and a Col-projection layer $f_p$. For an input image $\mathbf{X}$, the vision encoder computes a sequence of visual embeddings $f_v(\mathbf{X})$, which are then concatenated with token embeddings $\mathbf{y}_v$ derived from a fixed text prompt: "\nDescribe the image." This combined input is fed into the LLM. The projection layer $f_p$ then transforms the LLM's last hidden states into a low-dimensional feature space, resulting in feature sequences that can be represented as $\mathbf{E}_v = f_p(f_l^r(f_v(\mathbf{X}), \mathbf{y}_v))$. Similarly, for an input question $q$, the question is first augmented into $y_q$ using a prompt template. Then, its token embedding $\mathbf{y}_q$ is processed without visual input as $\mathbf{E}_q = f_p(f_l^r(\mathbf{y}_q))$. Finally, a late-interaction score $s_{\mathrm{LI}}(\mathbf{E}_q, \mathbf{E}_v)$ is computed between the feature sequences, measuring the relevance of a page image to the question text. More details about scoring method is provided in section 3.2.

**Question-Answering Module**  The QA module uses a classic LLaVA-like architecture (Liu et al., 2024b), utilizing a vision encoder $f_v$ to compute visual embeddings, which are combined with token embeddings and processed by an LLM $f_l^a$. The LLM then generates text answers autoregressively through next-word prediction.

## 3.2 Contextualized Late Interaction

We utilize the contextualized late interaction (Col) technique (Khattab & Zaharia, 2020) to compute relevance scores for evidence retrieval. Unlike traditional single-vector encoders, such as CLIP (Radford et al., 2021), the Col technique introduces an inter-sequence similarity metric called the late-interaction score, which captures more fine-grained question-image relevance. Formally, the late-interaction score between a text feature sequence $\mathbf{E}_q = \{\mathbf{e}_{q_1}, \ldots, \mathbf{e}_{q_n}\}$ of length $n$ and a visual feature sequence $\mathbf{E}_v = \{\mathbf{e}_{v_1}, \ldots, \mathbf{e}_{v_m}\}$ of length $m$ is defined as:

$$s_{\mathrm{LI}}(\mathbf{E}_q, \mathbf{E}_v) = \sum_{i=1}^{n} \max_{j \in \{1,..,m\}} \mathbf{e}_{q_i} \cdot \mathbf{e}_{v_j}^T. \tag{1}$$

We use it as a similarity score in contrastive learning to facilitate ranked retrieval. Specifically, we train our retrieval module to maximize the late-interaction score between a question and its corresponding evidence image, considering these as positive pairs. We then identify the most similar, but unassociated, image within the batch to form the hardest negative pair and minimize the score for this pair. Figure A.1 shows a training pair example. The loss function is defined as:

$$\mathcal{L} = \log(1 + \exp(s_{\mathrm{LI}}(\mathbf{E}_q, \mathbf{E}_v^-) - s_{\mathrm{LI}}(\mathbf{E}_q, \mathbf{E}_v^+))). \tag{2}$$

The training process of the Col-retrieval module is summarized in Algorithm 1.

## 3.3 Parameter Sharing via Dual-Adapter

To reduce memory usage, we optimize the model by sharing a single MLLM that includes both the vision encoder $f_v$ and the language model $f_l$ across both the retrieval and QA modules. To

---

**Algorithm 1** Col-retrieval training

---

**Require:** Pre-trained MLLM $\{f_v, f_l^r\}$, training batch of evidence image and question pairs $\{(\mathbf{X}_1, \mathbf{y}_1), \cdots, (\mathbf{X}_b, \mathbf{y}_b)\}$.

1: Initialize the Col-projection layer $f_p$.
2: **while** not converged **do**
3:     Get $\mathbf{E}_v^i = f_p(f_l^r(f_v(\mathbf{X}_i), \mathbf{y}_i))$, $i \in \{1, ..., b\}$.
4:     Get $\mathbf{E}_q^i = f_p(f_l^r(\mathbf{y}_i))$, $i \in \{1, ..., b\}$.
5:     Compute $\mathbf{S}_{i,j} = s_{\text{LI}}(\mathbf{E}_q^i, \mathbf{E}_v^j)$.
6:     Get negative image index $\hat{i}$ for each $\mathbf{y}_i$: $\hat{i} = \arg\max_{j \in \{1,...,b\}, j \neq i}(\mathbf{S}_{i,j})$
7:     Gradient update using loss function Eq.(2),
      where $\mathbf{E}_v^{j-} = \mathbf{E}_{\hat{i}}$.
8: **end while**

---

accommodate the different tasks required by each module, we insert two sets of adapters into the $f_l$ using the LoRA method (Hu et al., 2021). In the retrieval module, we use a set of adapters $\Delta\mathbf{W}_r$ to create the retrieval-LLM, $f_l^r$. For the QA module, a different set of adapters $\Delta\mathbf{W}_a$ is added to the $f_l$, creating the QA-LLM, $f_l^r$. In this way, we support both tasks using a single LLM and vision encoder, adding only $\sim 2\%$ additional parameters.

## 4 VISR-BENCH

**Visually-rich Document Selection** About 4,000 PDF documents are crawled from the Web and contents of these documents are extracted via a document parser[1]. We keep the document with figures and throw away text-only or scan documents. To select documents with specific types of figures, we build a figure scheme that includes 19 figure types after reviewing different documents. We find some types of figures are not informative, such as logo and banner. We use the pretrained CLIP model `ViT-L/14-336` (Radford et al., 2021) to perform a figure classification on the extracted figures and keep 6 out of 19 types of figures, including tables, maps, diagrams, infographics, data charts, workflows, and screenshots. After that, we also annotate the document types for all selected documents.

**Question-Answer Collection** Document parser returns all document elements in JSON format and the figures are saved separately as image files. We retrieve the JSON file for the document to obtain the contexts of each figure. Then we combine the figures with their contexts and use GPT-4o (API version 2024-02-15-preview) to generate QA pairs. For the GPT-4o prompts, we provide two demonstrations and ask GPT-4o to generate a QA pair. In addition, we perform automatic verification using GPT-4o to ensure the quality of the generation. Specifically, we only provide the figure to GPT-4o and ask it with the generated question; if GPT-4o can answer it correctly, we will keep that QA pair in the SV-RAG Bench. This heuristic filter ensures that the answers are from document figures and double-checks the correctness of generated answers.

**Dataset Statistics** VisR-Bench contains 226 documents and 471 human-verified question-answer pairs. Figure 3 shows the distributions of the document types and the length distribution by document type. VisR-Bench has a great diversity of documents compared to previous work (Tanaka et al., 2023; Islam et al., 2023; Ma et al., 2024d).

## 5 EXPERIMENTS

We assess the performance of SV-RAG in evidence page retrieval and visual question answering capabilities. We first evaluate the retrieval accuracy of the Col-retrieval module within SV-RAG and compare it with several baselines on SlideVQA (Tanaka et al., 2023), MM-LongBench (Ma et al., 2024d), DUDE (Van Landeghem et al., 2023), DocVQA (Mathew et al., 2020; 2021) and VisR-Bench. We then conduct experiments on question answering using SV-RAG and compare the results with other LMM baselines, inlcuding single-page and cross-page VQA. All experiments are implemented

---

[1]Adobe Extract API: https://developer.adobe.com/document-services/apis/pdf-extract/

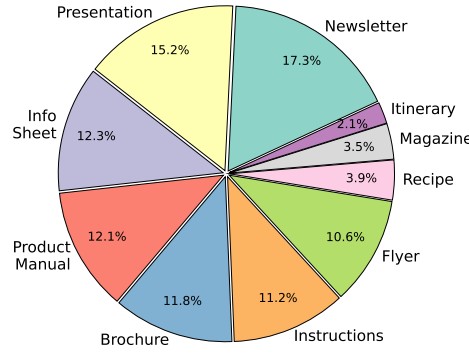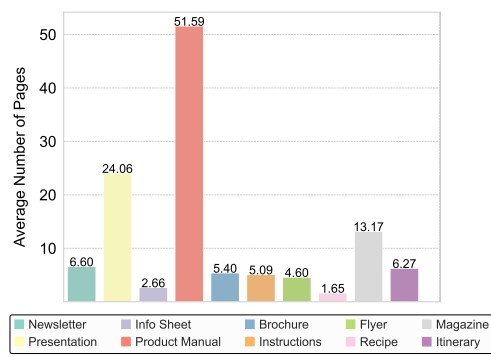

Figure 3: Distribution of document types (left) and average document lengths in each types (right).

with PyTorch and conducted on Nvidia A100 GPUs. The Col-retrieval modules are fine-tuned for 4 epochs with a batch size of 32 and a learning rate of 5e-5, using the AdamW optimizer and LoRA adapters on all linear layers in the LLM. The LoRA rank is set to 32.

## 5.1 DATASETS

**Finetuning Dataset**  We train our Col-retrieval modules using the original training data of ColPali (Faysse et al., 2024), which includes 39,463, 10,074, 13,251, 10,000, and 45,940 question-page pairs filtered from DocVQA, InfoVQA (Mathew et al., 2022), TATDQA (Zhu et al., 2024), arXivQA (Li et al., 2024), and synthetic data across various topics, including government reports, healthcare, artificial intelligence, and energy. We incorporated DocMatix-IR (Ma et al., 2024b) and PFL-DocVQA (Tito et al., 2023b), using GPT-4o to filter out duplicate images and unsuitable questions. The expanded dataset improved top-1 retrieval accuracy on MMLongBench-Doc by ∼1% without affecting other benchmarks. We fine-tuned our QA modules using the training split of the SlideVQA dataset (Tanaka et al., 2023). The SlideVQA dataset contains 1,919 slides in the training set, 400 in the test set, and 300 in the development set, with each slide consisting of 20 pages. The training split includes 10,290 samples, each annotated with questions, answers, and corresponding evidence.

**Evaluation Dataset**  We evaluated our method's performance on four public datasets—SlideVQA, MMLongBench-Doc (Ma et al., 2024d), DocVQA (Mathew et al., 2021), and DUDE (Van Landeghem et al., 2023)—along with our proposed VisR-Bench dataset. The evaluation was conducted in both single-evidence (SP) and cross-evidence (MP) settings, where questions require information from either a single page or multiple pages within a long document. For DocVQA, we used 5,349 SP and 5,187 MP QA pairs from the validation split. Similarly, we combined the test and dev splits of SlideVQA to form 2,995 SP and 763 MP QA pairs for evaluation. For DUDE, we evaluated 6,307 QA pairs from the validation split.

MMLongBench-Doc, which consists of 135 PDF documents averaging 50.4 pages (ranging from 9 to 468 pages), contains 1,081 QAs in total. From these, we extracted 488 single-evidence QAs to assess the performance of MLLMs designed for single-image tasks. Additionally, we report the results of our best-performing model across all categories in MMLongBench-Doc, providing a comprehensive comparison against state-of-the-art LMMs.

## 5.2 EVALUATION METRICS

We evaluate our model's performance on evidence retrieval and question-answering using several key metrics: Top-k Accuracy, Exact Match (EM) (Tanaka et al., 2023), Generalized Accuracy (G-Acc) (Ma et al., 2024d), Average Normalized Levenshtein Similarity (ANLS) (Biten et al., 2019), and Partial Normalized Levenshtein Similarity (PNLS) (Chen et al., 2024b). A detailed explanation of each metric can be found in Appendix B.

## 5.3 COMPARATIVE RETRIEVAL ACCURACY ANALYSIS

We evaluated the accuracy of the Col-retrieval module in SlideVQA, MMLongBench-Doc, SP-DocVQA, and VisR-Bench, comparing it with the baseline methods including CLIP (ViT-L/14) (Radford et al., 2021), SigLip (so400m-patch14-384) Zhai et al. (2023), BM25 (Robertson et al., 2009), SBERT (Reimers & Gurevych, 2019), BGE-M3 Chen et al. (2024c), BGE-large Xiao et al. (2023), and NV-Embed-v2 Lee et al. (2024). For encoder models, we used their text and image encoders to compute the cosine similarity between the feature of the question and the page. For text-based methods, the text content in the MMLongBench-Doc and SV-RAG Bench datasets is extracted using a document parser to ensure higher accuracy. For SlideVQA and SP-DocVQA, where only scanned images are available, the text is extracted using Paddle-OCR[2].

| accuracy | SlideVQA | | MMLong | | VisR-B | | SP-DocVQA | |
|---|---|---|---|---|---|---|---|---|
| | top1 | top5 | top1 | top5 | top1 | top5 | top1 | top5 |
| *Text-based Methods* | | | | | | | | |
| BM25 | 69.3 | 91.1 | 25.3 | 47.6 | 32.2 | 57.5 | 30.9 | 61.7 |
| SBERT | 73.0 | 91.0 | 44.7 | 70.2 | 38.8 | 72.1 | 47.4 | 74.0 |
| BGE-M3 | 74.3 | 92.0 | 42.7 | 66.6 | 47.7 | 78.1 | 47.8 | 77.5 |
| Bge-large | 81.3 | 93.3 | 47.4 | 71.5 | 53.7 | 80.3 | 56.7 | 81.5 |
| NV-Embed-v2 | 82.2 | 94.3 | 47.4 | 69.0 | 55.2 | 82.7 | 51.7 | 80.2 |
| *Encoder Models* | | | | | | | | |
| CLIP | 58.4 | 86.9 | 32.4 | 63.4 | 33.4 | 62.1 | 37.1 | 69.4 |
| SigLip | 66.2 | 90.1 | 44.9 | 69.4 | 53.2 | 81.3 | 39.3 | 71.9 |
| *Col-Retrieval Modules* | | | | | | | | |
| Col-Paligemma | 89.0 | 98.7 | 60.7 | 82.0 | 67.9 | 90.8 | 62.3 | 85.9 |
| Col-InternVL2 | 88.5 | 98.3 | 61.3 | 83.0 | 69.3 | 90.7 | 63.2 | 85.9 |
| Col-Phi-3-vision | 90.6 | 98.8 | 64.8 | 84.8 | 71.9 | 91.8 | 65.1 | 87.0 |

Table 1: Retrieval accuracy results on four datasets, where MMLong refers to MMLongBench-Doc, SV-RAG-B refers to SV-RAG-Bench. Bold font indicates the best model.

The results indicate that Col-retrieval outperforms all baselines, achieving more than 98% in top-5 retrieval accuracy on the SlideVQA dataset, where each slide consists of 20 pages. However, performance decreases on other datasets as the data become more complex and document lengths increase significantly.

## 5.4 MAIN RESULTS

We compared the performance of our method with popular lightweight LMMs on document question answering tasks, using PaliGemma (Beyer et al., 2024), Phi-3-v (Abdin et al., 2024), and InternVL2-4B (Chen et al., 2023b) as the backbone LMMs for both retrieval and QA modules, following the dual adapter design from Section 3.3. We fine-tuned the retrieval module using the 118,695 training question-page pairs used in ColPali (Faysse et al., 2024). The QA module is fine-tuned using SlideVQA's training split. We reported the original evaluation metrics used in prior works, including EM, G-Acc, and ANLS, and additionally reported PNLS, which better evaluates LLM-generated responses.

Table 2 presents the comparison results. We first evaluate SV-RAG on single-evidence questions from SP-SlideVQA, MMLongBench-Doc, and SP-DocVQA, where the required information is on a single page. To demonstrate the question-answering capabilities of LMMs, we include four "cheating" baselines where models are given the ground truth evidence page. Next, we test SV-RAG on cross-evidence questions from MP-SlideVQA, MP-DocVQA, and DUDE, where information spans multiple pages. We only test SV-RAG with InternVL2-4B backbone, since the other two LMM are pre-trained for single-page understanding. SV-RAG's performance is compared with classical encoder-only and encoder-decoder models, including BERT (Kenton & Toutanova, 2019), Longformer (Beltagy et al., 2020), Big Bird (Zaheer et al., 2020), T5 (Raffel et al., 2020), Hi-VT5 (Tito et al., 2023a), and LayoutLMv3 (Huang et al., 2022), with results taken from the best settings in the original

---

[2]PaddleOCR: https://github.com/PaddlePaddle/PaddleOCR

papers. InternVL2-8B and GPT-4o, processing all pages, serve as the state-of-the-art baselines for open-source and proprietary multipage LMMs, respectively. We demonstrate how the limitations of the retrieval and QA modules can impact overall performance through challenging examples from the SlideVQA dataset, as shown in Appendix C. Additional comparisons with text-only baselines utilizing a document parser are provided in Appendix G.1.

Table 2: **Quantitative Results in Multi-Page QA**: "#Param" refers to number of parameters. "Evidence" reports evidence setting: T (true evidence page), A (all pages), and Rk (top-k retrieved). Reported metrics include PNLS, Exact Match, Generalized Accuracy, and ANLS. † indicates models with LoRA adapter on QA module. Results for all encoder/decoder models are taken from their respective papers, with "-" indicating missing or not applicable results. Bold font indicates the best open-source model, excluding cheating baselines.

| Method | #Param | Evidence | SP-SlideVQA | | MMLongBench | | SP-DocVQA | |
|---|---|---|---|---|---|---|---|---|
| | | | EM | PNLS | G-Acc | PNLS | ANLS | PNLS |
| *Single-Page Evidence* | | | | | | | | |
| *Cheating Baselines* | | | | | | | | |
| PaliGemma | 3B | T | 37.30 | 0.63 | 23.9 | 0.38 | 0.65 | 0.79 |
| Phi-3-v | 4B | T | 13.72 | 0.80 | 33.7 | 0.52 | 0.65 | 0.85 |
| InternVL2 | 4B | T | 15.03 | 0.58 | 40.4 | 0.55 | 0.84 | 0.88 |
| GPT-4o | - | T | 30.59 | 0.84 | 56.8 | 0.62 | 0.87 | 0.94 |
| *Multi-image MLLMs* | | | | | | | | |
| InternVL2 | 8B | A | 12.62 | 0.65 | 14.1 | 0.22 | 0.50 | 0.55 |
| GPT-4o | - | A | 27.28 | 0.81 | 54.5 | 0.57 | 0.69 | 0.80 |
| *SV-RAG Models (Proposed)* | | | | | | | | |
| SV-RAG-PaliGemma | 3B | R1 | 35.03 | 0.60 | 23.9 | 0.35 | 0.56 | 0.69 |
| SV-RAG-PaliGemma† | 3B | R1 | 49.75 | 0.65 | 23.1 | 0.38 | 0.56 | 0.68 |
| SV-RAG-Phi-3-vision | 4B | R1 | 12.85 | **0.78** | 30.7 | **0.50** | 0.55 | 0.75 |
| SV-RAG-Phi-3-vision† | 4B | R1 | **58.13** | 0.77 | 28.4 | 0.44 | 0.68 | 0.73 |
| SV-RAG-InternVL2 | 4B | R5 | 16.40 | 0.58 | 33.2 | 0.48 | 0.70 | **0.76** |
| SV-RAG-InternVL2† | 4B | R5 | 45.07 | 0.77 | **34.0** | 0.49 | **0.71** | 0.75 |
| *Cross-Page Evidence* | | | | | | | | |

| Method | #Param | Evidence | MP-SlideVQA | | MP-DocVQA | | DUDE | |
|---|---|---|---|---|---|---|---|---|
| | | | EM | PNLS | ANLS | PNLS | ANLS | PNLS |
| *Encoder/Decoder models* | | | | | | | | |
| BERT-Large | 334M | - | - | - | 0.53 | - | 0.25 | - |
| Longformer | 148M | - | - | - | 0.55 | - | 0.27 | - |
| Big Bird | 131M | - | - | - | 0.58 | - | 0.26 | - |
| T5-Base | 223M | - | - | - | 0.51 | - | 0.42 | - |
| LayoutLMv3 | 125M | - | - | - | 0.55 | - | 0.20 | - |
| Hi-VT5 | 316M | - | - | - | 0.62 | - | 0.23 | - |
| *Multi-image MLLMs* | | | | | | | | |
| InternVL2 | 8B | A | 17.04 | 0.53 | 0.68 | 0.75 | 0.37 | 0.56 |
| GPT-4o | - | A | 16.09 | 0.73 | 0.67 | 0.79 | 0.54 | 0.70 |
| *SV-RAG Models (Proposed)* | | | | | | | | |
| SV-RAG-InternVL2 | 4B | R5 | 24.25 | **0.61** | 0.70 | **0.76** | 0.36 | **0.57** |
| SV-RAG-InternVL2† | 4B | R5 | **31.98** | 0.59 | **0.71** | 0.76 | **0.45** | 0.54 |

**Retrieval vs Multipage** We observe SV-RAG consistently outperforms InternVL2-8B, across various settings. The primary issue with LMMs is that long documents are transformed into excessively long visual token sequences, leading to significant memory burdens, as reported later in section 5.5. In datasets like MMLongBench-Doc and DocVQA, some documents exceed hundreds of pages, causing out-of-memory errors, even on servers with $8 \times$ A100 (80GB) GPUs. In such cases, we assigned a zero score in our experiments. In contrast, GPT-4o exhibits strong multi-page reasoning capabilities. However, the accuracy of the cheating baseline slightly surpasses that of using all pages, as providing only the evidence pages helps GPT-4o avoid distractions from irrelevant information in the longer context. Moreover, SV-RAG with InternVL2-4B backbone perform slightly better than the one with Phi-3-vision backbone on MMLongBench-Doc and SP-DocVQA, possibly due to the improvement in retrieval accuracy by using top-5 pages, which is more crucial for longer documents.

**Impact of Fine-tuning** We observe that SV-RAG QA modules with PaliGemma and InternVL2-4B backbones show a significant increase in EM on the SlideVQA dataset, surpassing their cheating baselines after fine-tuning on SlideVQA. The model with the Phi-3-vision backbone shows notable improvements in Exact Match (EM) scores without gains in PNLS, suggesting that fine-tuning primarily enhanced the model's attention and answer formatting. This could be because the pre-trained model was already optimized for these question types. Nevertheless, as shown in Figure D.1, we empirically find that fine-tuning still improves answering performance. However, we notice a performance drop for fine-tuned SV-RAG-Phi-3-vision on MMLongBench-Doc, indicating that fine-tuning can harm LLM generalization. A similar trend is seen with the InternVL2-4B backbone on the DUDE dataset.

**Comparison with SOTA LMMs** Finally, we present the complete results of SV-RAG-InternVL2-4B on the MMLongBench-Doc dataset to highlight the advantages of our method. As shown in Table 3, our model, with only 4 billion parameters, outperforms all open-source LMMs and achieves performance comparable to proprietary models such as Claude-3 Opus and Gemini-1.5-Pro.

| Method | #Param | Evidence Source | | | | | Evidence Page | | | ACC | F1 |
|---|---|---|---|---|---|---|---|---|---|---|---|
| | | TXT | LAY | CHA | TAB | FIG | SIN | MUL | UNA | | |
| *Open-source Models* | | | | | | | | | | | |
| DeepSeek-VL-Chat | 7.3B | 7.2 | 6.5 | 1.6 | 5.2 | 7.6 | 5.2 | 7.0 | 12.8 | 7.4 | 5.4 |
| Idefics2 | 8B | 9.0 | 10.6 | 4.8 | 4.1 | 8.7 | 7.7 | 7.2 | 5.0 | 7.0 | 6.8 |
| MiniCPM-Llama3-V2.5 | 8B | 11.9 | 10.8 | 5.1 | 5.9 | 12.2 | 9.5 | 9.5 | 4.5 | 8.5 | 8.6 |
| InternLM-XC2-4KHD | 8B | 9.9 | 14.3 | 7.7 | 6.3 | 13.0 | 12.6 | 7.6 | 9.6 | 10.3 | 9.8 |
| mPLUG-DocOwl 1.5 | 8.1B | 8.2 | 8.4 | 2.0 | 3.4 | 9.9 | 7.4 | 6.4 | 6.2 | 6.9 | 6.3 |
| Qwen-VL-Chat | 9.6B | 5.5 | 9.0 | 5.4 | 2.2 | 6.9 | 5.2 | 7.1 | 6.2 | 6.1 | 5.4 |
| Monkey-Chat | 9.8B | 6.8 | 7.2 | 3.6 | 6.7 | 9.4 | 6.6 | 6.2 | 6.2 | 6.2 | 5.6 |
| CogVLM2-LLaMA3-Chat | 19B | 3.7 | 2.7 | 6.0 | 3.2 | 6.9 | 3.9 | 5.3 | 3.7 | 4.4 | 4.0 |
| InternVL-Chat-v1.5 | 26B | 14.0 | 16.2 | 7.1 | 10.1 | 16.6 | 14.9 | 12.2 | **17.5** | 14.6 | 13.0 |
| EMU2-Chat | 37B | 6.1 | 9.7 | 2.6 | 3.8 | 7.7 | 5.7 | 6.1 | 16.5 | 8.3 | 5.5 |
| *SV-RAG Models (Proposed)* | | | | | | | | | | | |
| SV-RAG-InternVL2 (R5) | 4B | **26.5** | 18.8 | 22.3 | 19.6 | 23.6 | 33.2 | **13.1** | 12.4 | 22.2 | 22.8 |
| SV-RAG-InternVL2[†] (R5) | 4B | 26.3 | **22.1** | **25.0** | **20.7** | **25.2** | **34.0** | 10.6 | 15.7 | **23.0** | **24.2** |
| *Proprietary Models* | | | | | | | | | | | |
| Claude-3 Opus | - | 24.9 | 24.7 | 14.8 | 13.0 | 17.1 | 25.6 | 13.8 | 7.6 | 17.4 | 18.1 |
| Gemini-1.5-Pro | - | 21.0 | 17.6 | 6.9 | 14.5 | 15.2 | 21.1 | 11.1 | 69.2 | 28.2 | 20.6 |
| GPT-4V | - | 34.4 | 28.3 | 28.2 | 32.4 | 26.8 | 36.4 | 27.0 | 31.2 | 32.4 | 31.2 |
| GPT-4o | - | 46.3 | 46.0 | 45.3 | 50.0 | 44.1 | 54.5 | 41.5 | 20.2 | 42.8 | 44.9 |

Table 3: **Performance of various models on MMLongBench-Doc.** Questions are categorized in two ways: (1) by evidence source type—text (TXT), layout (LAY), chart (CHA), table (TAB), and image (IMG); and (2) by evidence pages—single-page (SIN), cross-page (MUL), and unanswerable (UNA). Models using LoRA adapters fine-tuned on SlideVQA for the QA module are marked with †. Bold font indicates the best open-source model. The results of baseline models are adopted from the original MMLongBench-Doc paper Ma et al. (2024d).

## 5.5 EFFICIENCY OF DIFFERENT MODELS

To evaluate the efficiency of SV-RAG, we conducted experiments on the SlideVQA dataset, which has 20 pages per question with a resolution of 1024x768. We recorded peak GPU memory usage and time costs for retrieval and QA modules separately. The GPU memory is manually recorded using the `nvidia-smi` command, which tends to report higher numbers than the actual memory required by the application due to overhead and memory management processes. We tested backbones including PaliGemma, Phi-3-v, and InternVL2-4B, all equipped with LoRA adapters. Since PaliGemma and Phi-3-v are single-page models, we used top-1 retrieved image as input. InternVL2-4B, however, supports multi-image input, allowing us to test with the top-1, 5, and 12 retrieved images.

As shown in Table 4, the QA module's memory consumption increases with the number of evidence pages, with 13 images (1024x768) exceeding the 80GB limit on an A100 GPU resulting in out-of-memory error. In contrast, the retrieval module maintains low memory usage, as SV-RAG processes pages independently, with costs equivalent to single-page reasoning. Although multi-evidence QA

| SV-RAG-Backbone | Page | Retrieval | | QA | |
|---|---|---|---|---|---|
| | | Time | Mem | Time | Mem |
| Paligemma | R1 | 2.3 | 9.2 | 1.0 | 12.4 |
| Phi-3-vision | R1 | 4.1 | 11.6 | 0.9 | 12.9 |
| InternVL2-4B | R1 | 9.2 | 14.2 | 1.4 | 14.6 |
| InternVL2-4B | R5 | 9.2 | 14.2 | 2.8 | 40.8 |
| InternVL2-4B | R12 | 9.2 | 14.2 | 4.1 | 76.4 |

Table 4: Time (s) cost and Peak GPU memory (GB) cost of SV-RAG models with different backbones.

requires more memory, SV-RAG remains efficient and compact, making it well-suited for answering questions from fewer evidence pages in resource-constrained environments. This demonstrates SV-RAG's ability to balance performance and resource usage, ensuring scalability across diverse deployment scenarios. Additional results on the retrieval efficiency are presented in Table F.1.

## 5.6 ABLATION

In our experiment, we use the hidden states from the last transformer layer (index 31) as the feature sequence. However, LLMs consist of multiple transformer layers, each encoding different types of information. To assess the impact of layer selection, we conduct an ablation study on the hidden states used to compute the late interaction score in Eq.(1). Given the high computational cost of training the col-retrieval module across all layers, we instead evaluate top-1 accuracy on the MMLongBench-Doc dataset using hidden states from different layers of the Phi-3-vision model with pre-trained weights.

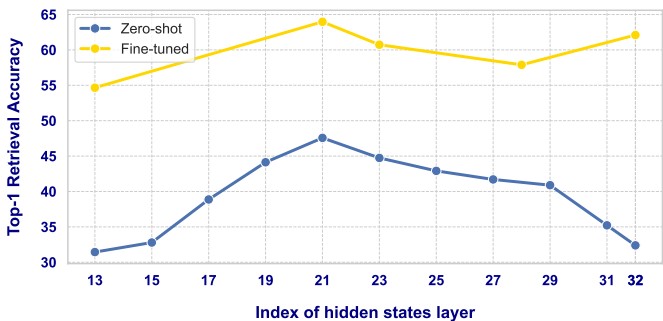

Figure 4: Top-1 retrieval accuracy on MMLongBench-Doc using different hidden states across all layers of Phi-3-vision.

Figure 4 shows that the hidden states of the 21th layer yield the highest accuracy. After fine-tuning a model with hidden states from this layer, we observed improved accuracy compared to using hidden states of the final layer. In particular, using hidden states from earlier layers can significantly reduce computational costs, enabling faster retrieval during inference.

## 6 CONCLUSIONS

In this paper, we propose SV-RAG, a lightweight MLLMs for visually-rich document understanding. SV-RAG has a unique design to facilitate multi-page document understanding using dual LoRA adapters. The research highlights that small open-source models are great at processing multipage documents and underscored the importance of efficient retrieval mechanisms in filtering irrelevant pages. Furthermore, we collect the VisR-bench dataset for document understanding, and empirical results on benchmarks demonstrated the effectiveness of SV-RAG. We hope these findings provide valuable insights for optimizing lightweight MLLMs, aiming to improve accuracy and efficiency in visually-rich document understanding.

## 7    LIMITATIONS

SV-RAG is the first MLLM that can perform visual-retrieval generation for document question answering using a single model. However, it still requires computational resources for training and inference, which may limit its practical applicability in resource-constrained environments. SV-RAG should be mobile friendly, as it only requires a single base model. This base model can be Phi-3-Silica within MS operating systems or an Apple on-device model within Apple IOS 18. A routing mechanism in Apple Intelligence can better balance computational cost and performance. However, our experiments are not performed on these real-world devices, which are necessary for pushing forward document intelligence.

## 8    ETHICS STATEMENT

The VisR-Bench dataset was curated with careful consideration of ethical and legal concerns. All documents are sourced from publicly available data with licenses explicitly permitting research use. To ensure data integrity and compliance, we provide links to the original sources instead of distributing the documents. Additionally, all QA pairs have been manually reviewed to exclude harmful content and personally identifiable information. The dataset does not expose sensitive user data, and experimental results are reported as aggregate statistics to prevent information leakage while ensuring reproducibility. These measures uphold ethical and legal standards while supporting responsible AI research.

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
