## A  EXAMPLE OF TRAINING PAIRS FOR RETRIEVAL MODULE

Figure A.1: Example of training pairs within a batch (batch size: 4) for contrastive training, using samples from the SlideVQA dataset.

## B  EVALUATION METRICS

We evaluate the model's performance on evidence retrieval and question-answering using five metrics explained as follows:

**Top-k Accuracy**   In our experiment, we focus on questions that have evidence from a single page. We use top-k accuracy to evaluate retrieval methods, which measures the percentage of times the evidence image appears within the top k most similar images.

**Exact Match**   Following (Tanaka et al., 2023), we report exact match (EM) frequency between generated answers and the ground truth, allowing for case insensitivity and extra spaces. While effective for fine-tuned models, this metric is less suited for LLM responses, which often include full sentences. Correct answers with extra context may thus be unfairly penalized.

**Generalized Accuracy**   We report generalized accuracy (G-Acc) from MMLongBench-Doc (Ma et al., 2024d), a GPT-dependent, rule-based evaluation protocol . Model responses are simplified using GPT-4o and scored based on answer-type-specific rules. However, G-Acc has two limitations: it introduces randomness from GPT's stochastic outputs and relies on answer-type annotations, limiting its applicability across datasets.

**ANLS**   Average Normalized Levenshtein Similarity (ANLS) (Biten et al., 2019) measures the similarity between predicted and ground truth text using the Levenshtein distance, normalized by the longer string's length. It outputs a similarity score between 0 and 1. ANLS allows mismatches, insertions, and deletions making it useful for OCR and document understanding tasks when exact matches are not required.

**PNLS**   The *partial normalized Levenshtein similarity* (PNLS) (Chen et al., 2024b) generalizes ANLS by not penalizing extra prefixes or suffixes while allowing mismatches, insertions, and deletions within the matched region. This makes it more suitable for evaluating LLM responses, which are often verbose to improve user experience.

The PNLS metric is formally defined as follows: String $\mathcal{T}_{1,m} = t_1 \ldots t_m$ represents the true answer and $\mathcal{S}_{1,n} = s_1 \ldots s_n$ is a model generated string. We first use using the approximate string matching algorithm (Sellers, 1980) to identify the sub-string of $\mathcal{S}$ that has the minimum edit distance to $\mathcal{T}$. Specifically, we first construct a scoring matrix $\mathbf{F}$ of size $(m+1) \times (n+1)$, where $F_{i,j}$ stores the smallest edit distance between the $i$-prefix $\mathcal{T}_{1,i}$ and any sub-string $\mathcal{S}_{x,j}, \forall x \in \{1, \ldots, j-1\}$ that ends at position $j$. The scoring matrix can be computed recursively

$$F_{i,j} = \begin{cases} 0 & \text{if } i = 0 \\ m & \text{if } j = 0 \\ \min \begin{pmatrix} F_{i-1,j-1} + c(t_i, s_j) \\ F_{i-1,j} + 1 \\ F_{i,j-1} + 1 \end{pmatrix} & \text{otherwise,} \end{cases}$$

where $c$ is the substitution cost that takes a value of 0 if $t_i = s_j$ and 1 otherwise. Once $\mathbf{F}$ is computed, the minimum value in the last row is the optimal edit distance and the end index of the matched sub-string $j' = \arg\min_j (F_{m+1,j})$. The start index $i'$ can be found by tracing back the the computation of Eq.(B) using $\arg\min$ operation. Finally, the PNLS is computed as: $m/(m + j' - i' + 1)$. In our experiments we use binary cost function: $c(t_i, s_j) = 0$ if $t_i = s_j$ else $c(t_i, s_j) = 1$

## C   EXAMPLE OF INFERENCE FAILURE SCENARIO

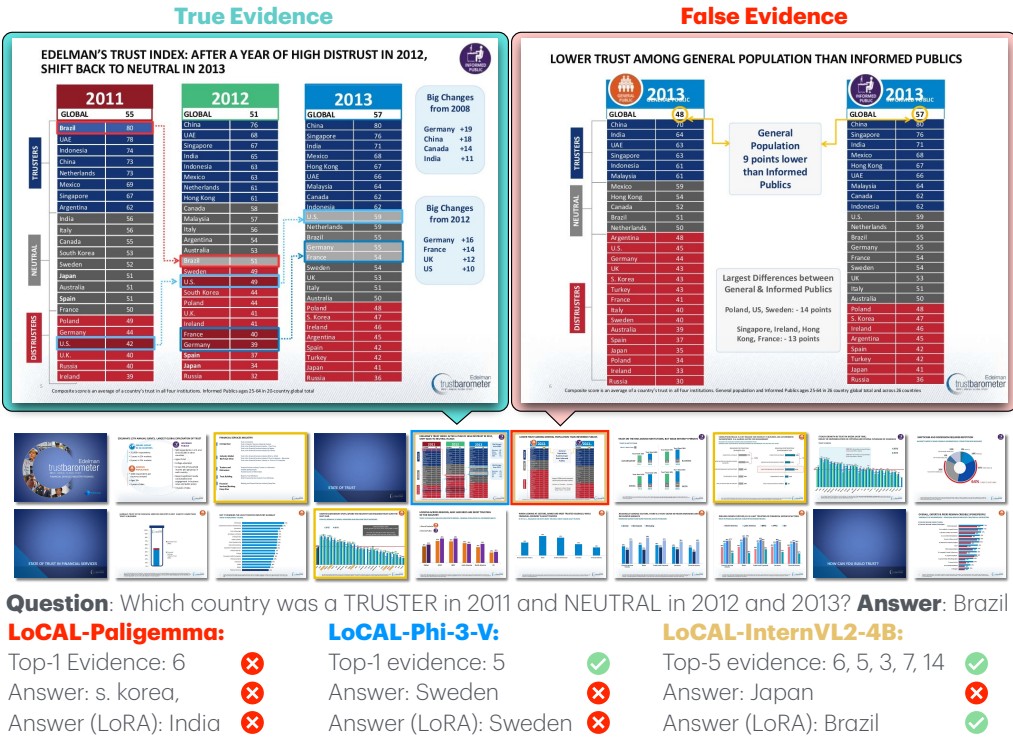

Figure C.1: Inference example of a challenging case in the SlideVQA dataset. SV-RAG-Paligemma retrieved the wrong evidence page due to limitations in its retrieval module, leading to an incorrect answer. SV-RAG-Phi-3-V retrieved the correct page but provided a wrong answer due to limitations in its QA module. Meanwhile, SV-RAG-InternVL2-4B also assigned the highest relevance score to an incorrect page. However, since it processes multiple pages (top 5), the correct evidence page was included in the input, allowing its fine-tuned QA module to deliver the correct answer.

## C.1 Additional Examples of Retrieval Failures

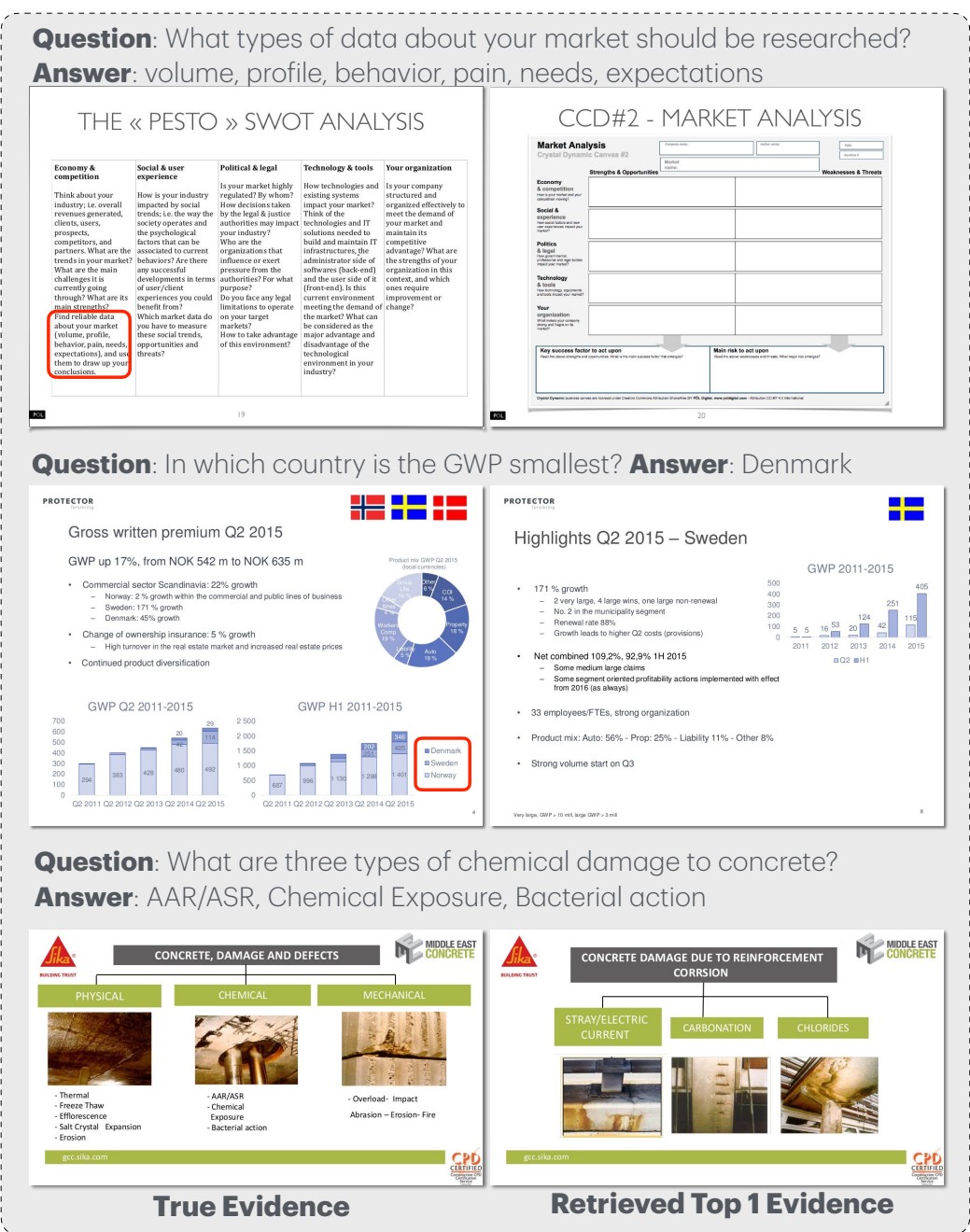

Figure C.2: Failure cases from the SlideVQA dataset, highlighting retrieval module errors. In the first two examples, some of the relevant information (highlighted in red boxes) on the true evidence pages is difficult even for human eyes to detect. In the third example, the retrieved page has a high similarity to the true evidence page, making it challenging to rank correctly. Additionally, answering the question accurately requires a deep understanding of the concept of chemical damage and related topics.

# D QAULITATIVE RESULTS IN QUESTION-ANSWERING

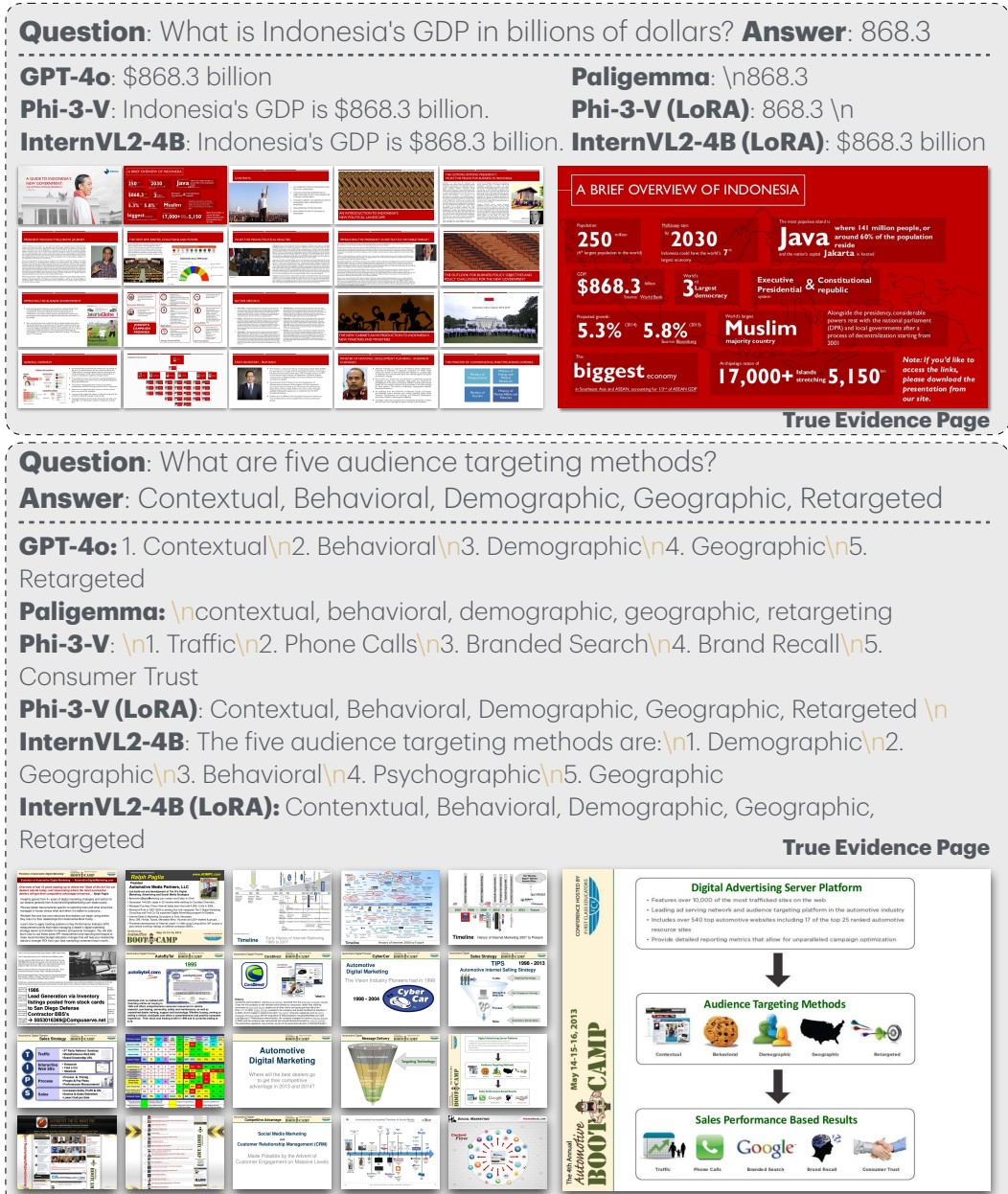

Figure D.1: Question answering examples on the SlideVQA dataset using different QA modules. Models without fine-tuning, such as Phi-3-V and InternVL2-4B, tend to produce verbose and error-prone responses. However, in the second example, fine-tuning with the LoRA adapter significantly improves the accuracy of Phi-3-V and InternVL2-4B.

# E   EXAMPLES FROM THE VISR-BENCH DATASET

Figure E.1: Example question-and-answer pair from the VisR-Bench dataset, highlighting the reliance on both image and surrounding text for accurate responses.

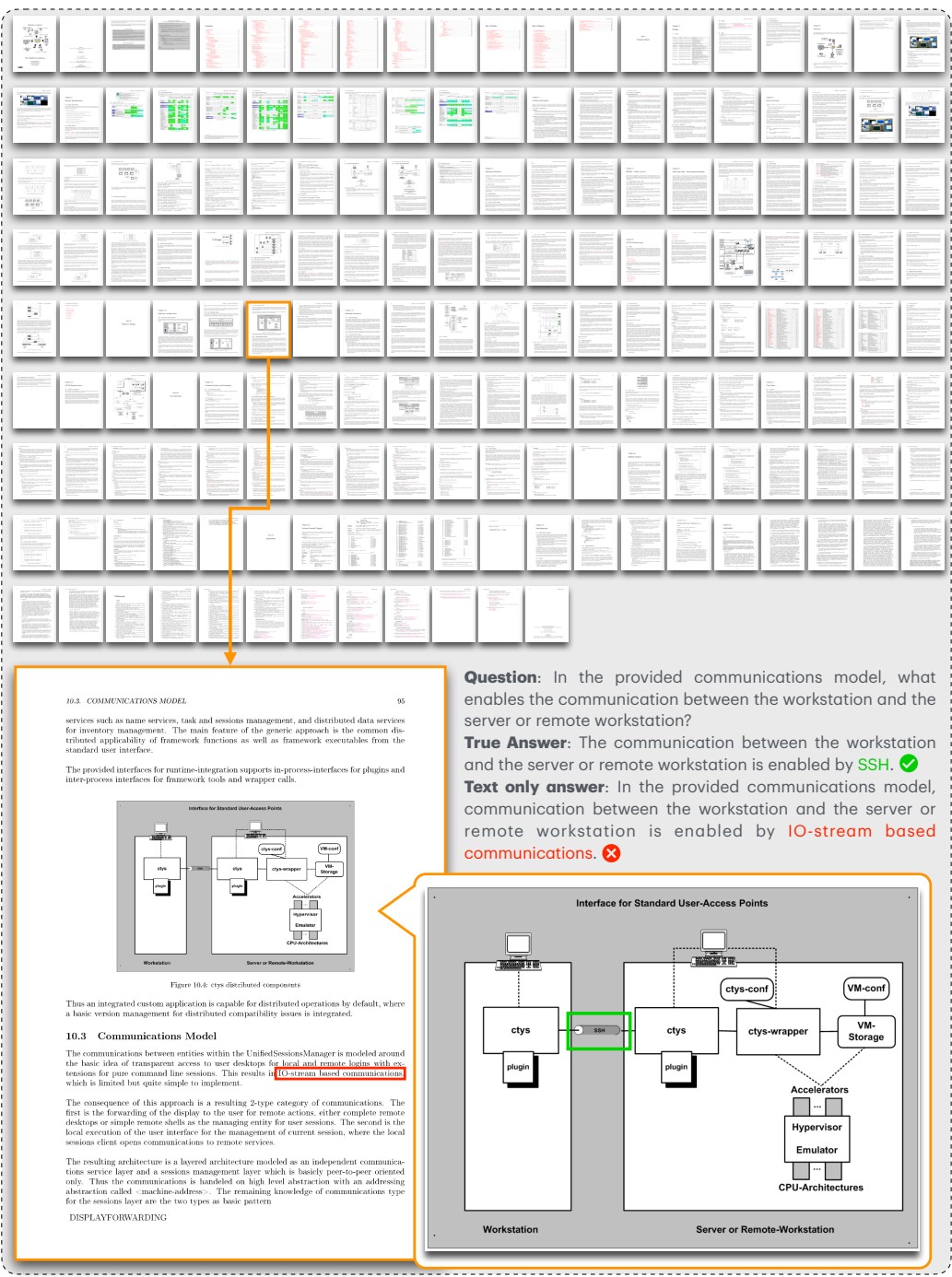

Figure E.2: Example question-and-answer pair from the VisR-Bench dataset, highlighting the reliance on both image and surrounding text for accurate responses.

## F    COMPARISON OF RETRIEVAL METHOD EFFICIENCY

| Text Extraction | | Text Encoding | | Multimodal Encoding | |
|---|---|---|---|---|---|
| PaddleOCR | 0.275 | BM25 | 0.0001 | CLIP | 0.022 |
| | | BGE-m3 | 0.131 | SigLip | 0.109 |
| PDF Parser | 0.762 | BGE-large | 0.137 | Col-Paligemma | 0.140 |
| | | NV-embed-v2 | 0.117 | Col-Phi-3-V | 0.230 |
| | | | | Col-InternVL2 | 0.581 |

Table F.1: Per-page time cost of retrieval methods: The left table presents time cost (seconds) of text-based methods that rely on text extraction techniques, such as OCR models, followed by text encoders to compute page embeddings. The right table presents time cost (seconds) of multi-modal methods that encode the entire page as an image.

## G    ADDITIONAL EXPERIMENT RESULTS

We compare our method with text-only baselines using a document parser[3] to highlight the advantages of MLLMs in multi-modal understanding. QA results are reported for the VisR-Bench and MMLongBench-Doc datasets, where PDF files are available.

Table G.1 presents QA results on VisR-Bench and MMLongBench-Doc datasets. To evaluate answer quality for VisR-Bench, where true answers are long and detailed, we introduce the Mean GPT Score (MGS), as string-matching methods often penalize variations in wording for lengthy answers. Instead, we prompt GPT-4o to compare a model's answer with the ground truth and assign a binary score based on detail alignment.

| QA Module | Retrieval Module | Evidence | VisR-B MGS | MMLong G-Acc |
|---|---|---|---|---|
| *Text only QA methods* | | | | |
| Phi-3 + parser | Col-Phi-3-V | R5 | 14.1 | 29.2 |
| GPT-4o + parser | Col-Phi-3-V | R5 | 24.9 | 43.2 |
| GPT-4o + parser | - | A | 27.6 | 42.4 |
| *MLLM QA models* | | | | |
| PaliGemma | Col-PaliGemma | R1 | 12.2 | 23.9 |
| Phi-3-V | Col-Phi-3-V | R1 | 24.2 | 30.7 |
| SV-RAG-InternVL2 | Col-InternVL2 | R5 | 25.2 | 33.2 |
| GPT-4o | Col-Phi-3-V | R5 | 47.2 | 55.1 |
| GPT-4o | - | A | 43.2 | 54.5 |

Table G.1: parser results

Our results indicate that using image evidence consistently outperforms text-only evidence. On VisR-Bench, text-only baselines showed a significant performance drop, emphasizing the dependency of questions on both image and text. However, the MGS of text-only baselines is not zero, likely because the model leverages text from a broader context rather than relying solely on the surrounding text, enabling it to extract relevant information even in the absence of visual input.

Additionally, reducing input pages with the retrieval module improved GPT-4o's performance with image evidence, aligning with the findings in Table 2. In contrast, retrieval did not enhance GPT-4o's performance on VisR-Bench in the text-only setting, likely because the evidence pages lacked sufficient information to fully address the questions. Including additional context in such cases might yield better results.

---

[3]Adobe Extract API: https://developer.adobe.com/document-services/apis/pdf-extract/