# OpenReview forum: "SV-RAG: LoRA-Contextualizing Adaptation of  MLLMs for Long Document Understanding"
_ICLR.cc/2025/Conference — ICLR 2025 Poster_

### Official Review · Reviewer_nKcG · 2024-10-20

**Soundness:** 3
**Presentation:** 3
**Contribution:** 3
**Rating:** 6
**Confidence:** 4

**Summary:**

The paper aims to solve the problem of long document understanding. It proposes a method based on the LoRA technique that uses latent embeddings from LLMs to perform evidence retrieval and question answering at same time. To demonstrate the effectiveness of the method, the paper introduces a new dataset, LoCAL-Bench, and conducts experiments using multiple benchmarks.

**Strengths:**

1.	The paper is well-motivated, and the experiments demonstrate that the proposed method effectively addresses the challenge of long document understanding.
2.	The structure of the LoCAL method is well-designed and compatible with multiple existing LLM models. Furthermore, sharing LLM parameters fully leverages their linguistic capabilities and enhances memory efficiency, thereby extending applicability. The proposed solution in the article is simple, effective, and insightful.
3.	The experiments are well-organized, and comparisons with multiple baselines, including LLM-based and non-LLM-based methods, are sound and the results convincing.
4.	The paper is well-written, fluent, and highly readable.

**Weaknesses:**

1.	The LoCAL method primarily aims to solve the problem of multi-page long document understanding. However, the LoCAL-Bench dataset does not appear to feature the characteristics of long document length, which makes it hard to examine the real performance of the method. An ideal dataset should include a greater number of document pages than existing benchmarks. The authors should clarify whether the LoCAL-Bench dataset is necessary to demonstrate the effectiveness of the proposed method.
2.	Although the method achieves comparable performance to proprietary models such as Claude-3 Opus and Gemini-1.5-Pro, these proprietary models still exhibit strong abilities in other downstream tasks. However, the generalization ability of the method remains untested, which is crucial for LLM-based methods. Additionally, some work such as Textmonkey (Yuliang Liu, Biao Yang, Qiang Liu, Zhang Li, Zhiyin Ma, Shuo Zhang, and Xiang Bai. Textmonkey: An ocr-free large multimodal model for understanding document. arXiv preprint arXiv:2403.04473, 2024d.) also test their performance on multiple downstream tasks. Therefore, the author should conduct the experiments on the generalization ability or explain why such experiments are unnecessary for the paper.
3.	The evidence pages supported by the proposed method seem to be limited to five pages. When the relevant evidence pages for a question exceed five, performance may decrease. This characteristic may reduce the method's general performance.

**Questions:**

See above.

---

> ### Author Response · Authors · 2024-11-27
>
> **W1**: The authors should clarify whether the LoCAL-Bench dataset is necessary to demonstrate the effectiveness of the proposed method.
>
> **A**: As described in our global response Q1, the primary motivation behind LoCAL-Bench is to demonstrate the multimodal understanding capabilities of models. The dataset includes questions designed to require the integration of both images and surrounding text for accurate answers, emphasizing multimodal reasoning rather than document length. Examples of such questions are provided in Appendix E.
>
> **W2**: The generalization ability of the method remains untested, and lack of discussion on its performance on downstream tasks compared to other works.
>
> **A**: We have cited TextMonkey in both the introduction and related work sections to provide context and acknowledge its contributions.
>
> Our primary contribution is not to surpass large proprietary models like Claude-3 Opus or Gemini-1.5-Pro in overall capabilities, such as generalization ability, as smaller models naturally face capacity limitations compared to these larger models. Instead, our framework focuses on leveraging the base LMM’s existing generalization ability to improve long-document QA performance. By integrating a retrieval mechanism, our approach reduces distracting context in long inputs, allowing the base LMM to perform better on long-document QA compared to processing all pages directly.
>
> Additionally, our QA module can be fine-tuned on specific QA datasets for enhanced task-specific performance or left unchanged to preserve the base LMM’s generalization ability. For instance, TextMonkey could also serve as the base LMM within our framework. The resulting LoCAL-TextMonkey model would enable scalable, efficient long-document QA while maintaining the original model’s generalization ability for single-page QA tasks.
>
>
>
> **W3**: The method’s performance may decline when relevant evidence pages exceed the five-page limit, potentially affecting its general performance.
>
> **A**: Thank you for highlighting this potential limitation of our framework. Our method is specifically designed to address non-summary QA tasks, where questions typically rely on localized information from a small number of pages. The choice to use the top 5 pages reflects a practical balance between 1. hardware constraints and 2. the requirements of widely used benchmarks:
>
> 1. As shown in Table 4, the memory cost of directly processing long documents remains a significant challenge for current LMM architectures, and our framework provides a practical and scalable solution to effectively handle long-document understanding within these limitations.
>
> 2. In the MMLongBench-Doc dataset, only 3.41% of questions require more than five evidence pages, and in the SlideVQA dataset, all questions require fewer than two evidence pages, suggesting that our framework is well-aligned with the requirements of these benchmarks and many real-world tasks.

---

### Official Review · Reviewer_inGb · 2024-10-29

**Soundness:** 3
**Presentation:** 3
**Contribution:** 2
**Rating:** 6
**Confidence:** 4

**Summary:**

This paper introduces the LoCAL framework, which aims to enhance the understanding of multi-page, visually-rich documents by large multimodal models (LMMs). The framework employs dual LMM adapters for evidence page retrieval and question answering, demonstrating state-of-the-art performance on public benchmarks. The proposed method involves using LMMs as multimodal retrievers to fetch relevant pages and answer user questions based on these pages, utilizing hidden states for efficient retrieval. The paper also introduces a new dataset, LoCAL-bench, comprising 226 documents and 471 question-answer pairs across nine domains. The results highlight the effectiveness of LoCAL.

**Strengths:**

1.The paper introduces the LoCAL framework, which effectively broadens the capabilities of large multimodal models (LMMs) for understanding multi-page, visually-rich documents. This is a significant advancement in the field.

2. The implementation of dual LMM adapters for evidence page retrieval and question answering is a novel approach that enhances the efficiency and performance of the models.

3. The introduction of the LoCAL-bench dataset, which includes a diverse range of documents and question-answer pairs, provides a valuable resource for further research and development in this area.

**Weaknesses:**

While the paper does mention the limitations of traditional methods using document parsers for retrieval-augmented generation, it focuses on introducing and evaluating the LoCAL framework without a direct comparison to RAG methods.

**Questions:**

How to evaluate the quality of the LoCAL-bench dataset? Because the data is purely genereted from the GPT-4o model, although with human verificaition I still doubt its quality compared with other human-labeld datasets.

**Details Of Ethics Concerns:**

The web PDF data used for labeling the LoCAL-bench may contain sensitive information and needs to be carefully reviewed for public benchmarking.

---

> ### Author Response · Authors · 2024-11-27
>
> **W**: The paper lacks a direct comparison between the LoCAL framework and Retrieval-Augmented Generation (RAG) methods using document parsers.
>
> **A**: Thank you for pointing this out. The document parser (Adobe Extraction API, cited in section 4) extracts text from PDF files while ignoring images, making it unsuitable for scanned document image datasets and limiting its capability to support multimodal understanding. Additionally, the parser incurs a computational overhead of approximately 10 seconds per call, further reducing its practicality.
> As noted in our global response to Q1, we provided experimental results on MMLongBench-Doc and LoCAL-Bench, where text-only LLMs, such as GPT-4 and Phi-3 (the language model used in Phi-3-V and InternVL2), were used to answer questions based on parser-extracted text from retrieved pages. These results showed lower performance compared to corresponding LMM methods that directly process pages as scanned images. We will include these results and provide further discussion in the revised manuscript.
>
> **Q**: How to evaluate the quality of the LoCAL-bench dataset
>
> **A**: As noted in the global response Q2, LoCAL-Bench is a small dataset of 226 QA pairs, carefully reviewed by human annotators to ensure accuracy and consistency. GPT-4o was used primarily to filter the initial large collection of documents, selecting content suitable for creating questions that require both image and surrounding text to answer. We have included two examples in Appendix E to illustrate the dataset’s characteristics. The small size of the dataset allows for thorough human inspection, ensuring its quality is comparable to human-annotated benchmarks.

---

### Official Review · Reviewer_yV7k · 2024-11-04

**Soundness:** 3
**Presentation:** 3
**Contribution:** 2
**Rating:** 6
**Confidence:** 4

**Summary:**

This paper introduces LoCAL (LoRA-Contextualizing Adaptation of Large multimodal models), a framework for extending LMMs to handle long, multi-page document understanding. The key insight is using LMMs themselves as efficient retrievers for relevant pages before performing question answering, rather than relying on traditional document parsers or trying to process entire documents at once. It implements efficient parameter sharing through dual LoRA adapters to build a dual-module architecture where a single LMM serves both as a retriever and question answerer.

**Strengths:**

- The paper introduces an approach to multipage document understanding by combining LLMs with a retrieval mechanism tailored for multi-page, visually-rich documents. The use of dual LoRA adapters for separate retrieval and question-answering tasks is a creative adaptation that enhances the efficiency and modularity of the model.
- The paper provides an extensive set of experiments across multiple datasets, including SlideVQA, MMLongBench-Doc, DocVQA, and the newly proposed LoCAL-bench.

**Weaknesses:**

The idea of using LMM for evidence page retrieval is interesting but not entirely novel. Previous work like ColBERT (Khattab & Zaharia, 2020) has already employed contextualized late interaction in document retrieval tasks. Moreover, using LORA to efficiently adapt LLMs for different tasks is also widely used in many scenarios. Therefore, the paper could better position its contribution by clearly delineating how LoCAL surpasses existing methods in LMM retrieval and adaptation for long documents.

**Questions:**

- Based on the fine-tuning dataset, you trained the retrieval module using the original ColPali training data, supplemented with DocMatix-IR and PFLDocVQA. Have you noticed any performance improvements compared to the retrieval module trained only on the original ColPali training data on Table 1 benchmarks and also ViDoRe benchmark from ColPali?
- According to the ColPali paper, the authors use the BGE-M3 embedding model as the text-based baseline. Do you believe this model could significantly outperform the SBERT baseline, given that BGE-M3 is more advanced on existing benchmarks?

**Details Of Ethics Concerns:**

The paper mentions web-crawling around 4,000 PDF documents from the web to build LoCAL-Bench. Given that these documents may be subject to copyright protection, there are potential legal and ethical issues related to the use of copyrighted materials.

---

> ### Author Response · Authors · 2024-11-27
>
> **W**: The paper could strengthen its contribution by clearly distinguishing how LoCAL improves upon existing methods like ColBERT and common LoRA-based adaptations in LMM retrieval and long-document understanding.
>
> **A**: Thank you for your valuable feedback. We are grateful for the opportunity to clarify our contributions.
> Our approach provides an effective solution for long-document QA by unifying retrieval and QA tasks within a single framework, achieved by customizing a Large Multimodal Model (LMM) with LoRA adapters. This design eliminates the reliance on external retrieval systems, allowing the LMM to function as both the retrieval and QA module. By focusing on the selection of relevant pages rather than processing all pages simultaneously, our method offers a scalable and efficient solution for multi-page document understanding, aligning closely with real-world use cases.
> In comparison to ColBERT, which is limited to text-only inputs, our approach natively processes multimodal inputs. This avoids the inefficiencies associated with lossy and time-consuming OCR processes, resulting in superior retrieval performance, as detailed in Appendix F.
>
> **Q1**: Has the addition of DocMatix-IR and PFLDocVQA to the original ColPali training data improved the retrieval module’s performance on Table 1 benchmarks and the ViDoRe benchmark?
>
> **A**: We trained two versions of our retrieval module with the PaliGemma backbone: one using only ColPali training data and another using combined data from additional sources (DocMatix-IR and PFLDocVQA). Using GPT4o with a tailored prompt, we filtered out duplicate images to minimize potential overfitting and excluded unsuitable questions, such as:
> 1. Broad questions: Requiring summarization beyond a single page.
> 2. Non-specific questions: Not tied to image content (e.g., “What is the page number?”).
> 3. Cross-page reasoning questions: Requiring information from multiple pages.
>
> The model trained with additional data achieved a ~1% improvement in top-1 accuracy on MMLongBench-Doc and a ~1.5% improvement in NDCG@5 on the ViDoRe benchmark, while maintaining comparable performance on other datasets. However, we were unable to reproduce the performance of ColPali 1.0 using released source code and training data on the ViDoRe leaderboard, possibly due to hyperparameter differences. Thus, we use the ColPali 1.0 checkpoint in our experiments as a stronger baseline.
>
> **Q2**: How does the performance compare between BGE-M3 and SBERT?
>
> **A**: Thank you for your question. We have reported the results for BGE-M3 in the global response Q3. Despite these additions, our method consistently outperforms all others, highlighting its strength. We have updated Table 1 of our manuscript to include these new results.

---

> > ### Comment · Reviewer_yV7k · 2024-12-02
> > **Thank you to the authors for your explanation.**
> >
> > Thank you to the authors for the response. Most of my concerns have been addressed, and I would like to maintain my positive rating.

---

### Official Review · Reviewer_WtC2 · 2024-11-04

**Soundness:** 3
**Presentation:** 3
**Contribution:** 3
**Rating:** 6
**Confidence:** 3

**Summary:**

It presents a framework named LoRA-Contextualizing Adaptation of Large multimodal models (LoCAL) to broaden the horizons of LMM for multi-page document understanding.
LoCAL is implemented with two specific LMM adapters, one for evidence page retrieval and the other for question answering.
Empirical results show state-of-the-art performance on public benchmarks, demonstrating the effectiveness of LoCAL.

**Strengths:**

1. A novel framework named LoCAL to broaden the horizons of LMMs, where it uses intermediate LMMs hidden embedding for efficient question-based evidence page retrieval.
2. It finetunes LMMs through dual LoRA adapters for evidence page retrieval and question answering.
3. It collects a visually-rich document QA dataset, LoCAL-bench.
4. It empirically demonstrates its effectiveness.

**Weaknesses:**

1. This article needs to be compared with more methods, such as bge-large, NV-Embed-v2, SigLIP, ColPali.
2. Two LoRA with one model, actually there are still two models, not a real unified model.

**Questions:**

See Weaknesses.

---

> ### Author Response · Authors · 2024-11-27
>
> **W1**: This article needs to be compared with more methods, such as bge-large, NV-Embed-v2, SigLIP, ColPali.
>
> **A**: Thank you for highlighting the missing methods in our evaluation. We have added the results for these methods in Table 1. Notably, ColPali is a specific instance of our framework. In Table 1, Col-Paligemma uses the same structure of ColPali and we used ColPali 1.0 checkpoint in this experiment. Despite these additions, our method consistently outperforms all others, further showcasing the advantage of our approach.
>
> **W2**: Two LoRA with one model, actually there are still two models, not a real unified model.
>
> **A**: Thank you for your valuable feedback. We acknowledge your point that using two LoRA adapters with a shared base model does not constitute a truly unified model.
>
> Our primary objective is to optimize GPU memory usage by sharing the base model between two LoRA adapters, rather than merging tasks into a single parameter set. Since retrieval and QA are distinct steps in our pipeline, using two specialized adapters provides a straightforward and efficient solution, aligning with established practices in foundation models, such as the Apple Intelligence Foundation Language Models [1], where adapters are used for various tasks. While a unified model trained with both contrastive and next-word prediction losses could eliminate the need for one set of LoRA adapters, it would introduce additional training complexities and potentially compromise performance. Furthermore, retrieval and QA still need to be performed as separate steps  to avoid the infeasible memory cost in processing more pages, as reported in Table 4.
>
> [1]. Gunter, Tom, et al. "Apple intelligence foundation language models." arXiv preprint arXiv:2407.21075 (2024).

---

### Author Response · Authors · 2024-11-27
**Global response**

We thank the reviewers for their positive feedback and valuable suggestions. We are pleased that they recognize the novelty and effectiveness of our framework. Below, we address common concerns regarding the proposed LoCAL Bench dataset and missing baselines. Specific questions raised by individual reviewers will be addressed separately. We will incorporate detailed revisions into the camera-ready version based on these responses.

---

> ### Author Response · Authors · 2024-11-27
>
> **Q1**: Is LoCAL Bench necessary to demonstrate the effectiveness of the proposed method.
>
> **A**: Thanks for the comments, which help us to realize the confusion in Section 4 regarding QA pair selection in LoCAL-Bench. We have included the updated description into the new version.
>
> LoCAL-bench shows the necessity of using large multimodal models for document understanding over text-based models that rely on document parsers to extract text. Specifically, questions within the LoCAL-bench are filtered to exclude questions answerable with textual information only. Hence, all questions require both figures and their surrounding texts from the document to answer. Compared with LoCAL-Bench, most existing benchmark questions can be answered using extracted text.
>
> We compare the QA performance of our method with text-only baselines that utilize the document parser on LoCAL-Bench and MMLongBench-Doc. We use GPT based evaluation as introduced in Section 5.2 and Appendix G. Our results show that multimodal models consistently outperform text-only baselines, with the gap being more pronounced on LoCAL-Bench, highlighting the dependency on both image and text. Using the retrieval module improved GPT-4o’s performance with image evidence. However, in the text-only setting, retrieval did not enhance GPT-4o’s performance, likely due to insufficient information in the evidence pages, where additional context could be beneficial.
>
> **Text-only QA**:
> | QA Module       | Retrieval Module| Evidence | LoCAL-B | MMLong |
> |-----------------|-----------------|----------|---------|--------|
> | Phi-3 + parser  | Col-Phi-3-V     | R5       | 14.1    | 29.2   |
> | GPT-4o + parser | Col-Phi-3-V     | R5       | 24.9    | 43.2   |
> | GPT-4o + parser | -               | A        | 27.6    | 42.4   |
>
>
> **Multi-Modal QA**:
> | QA Module       | Retrieval Module| Evidence | LoCAL-B | MMLong |
> |-----------------|-----------------|----------|---------|--------|
> | PaliGemma       | Col-PaliGemma   | R1       | 12.2    | 23.9   |
> | Phi-3-V         | Col-Phi-3-V     | R1       | 24.2    | 30.7   |
> | LoCAL-InternVL2 | Col-InternVL2   | R5       | 25.2    | 33.2   |
> | GPT-4o          | Col-Phi-3-V     | R5       | 47.2    | 55.1   |
> | GPT-4o          | -               | A        | 43.2    | 54.5   |
>
>
> **Q2**: Quality and Ethics Concerns of the LoCAL Bench dataset
>
> **A**: We thank the reviewers for highlighting this concern and would like to emphasize that creating synthetic datasets by crawling data from the web and leveraging models like GPT to generate datasets is a common practice in NLP research. LoCAL-Bench, derived from the web, has been curated and reduced to just 226 unique documents (as noted in section 4 data statistics). All QA pairs have been validated by human reviewers to ensure the exclusion of harmful contents and personal identifiable information. Additionally, we confirm that the licenses and usage terms of each document explicitly permit use for research purposes.
>
> To address ethical and legal concerns, the benchmark does not distribute the actual documents but instead provides links to their original sources, thereby avoiding the replication of real files while preserving dataset integrity. Furthermore, the experimental results are presented only as aggregate statistics, ensuring no potential information leakage. This approach guarantees reproducibility while strictly adhering to ethical and legal standards.

---

> ### Author Response · Authors · 2024-11-27
>
> **Q3**: Add more baselines for the retrieval task.
>
> **A**: We have added additional baselines for the retrieval task to provide a more comprehensive comparison. Furthermore, we have updated the CLIP results using the largest checkpoint and improved the results for our Col-Phi-3-V model with a more optimized checkpoint. Our method continues to achieve the best performance, further highlighting its advantages and robustness. These updates have been incorporated into the Table 1 of the revised manuscript.
>
> |             | SlideVQA |       | MMLong |       | LoCAL-B |       | SP-DocVQA |       |
> |-------------|----------|-------|--------|-------|---------|-------|-----------|-------|
> |             | Top 1    | Top 5 | Top 1  | Top 5 | Top 1   | Top 5 | Top 1     | Top 5 |
> | BGE-M3      | 74.3     | 92.0  | 42.7   | 66.6  | 47.7    | 78.1  | 47.8      | 77.5  |
> | Bge-large   | 81.3     | 93.3  | 47.4   | 71.5  | 53.7    | 80.3  | 56.7      | 81.5  |
> | NV-Embed-v2 | 82.2     | 94.3  | 47.4   | 69.0  | 55.2    | 82.7  | 51.7      | 80.2  |
> | CLIP        | 58.4     | 86.9  | 32.4   | 63.4  | 33.4    | 62.1  | 37.1      | 69.4  |
> | SigLIP      | 66.2     | 90.1  | 44.9   | 69.4  | 53.2    | 81.3  | 39.3      | 71.9  |
> | Col-Phi     | 90.6     | 98.8  | 64.8   | 84.8  | 71.9    | 91.8  | 65.1      | 87.0  |

---

### Meta-Review · Area_Chair_U61y · 2024-12-23

**Metareview:**

**Summary:**

The paper introduces LoCAL, a framework designed to enhance the ability of large multimodal models to understand long, visually rich documents. LoCAL addresses inefficiencies in traditional retrieval and direct processing methods by employing LMMs for evidence page retrieval and question answering, supported by LoRA adapters. The approach leverages hidden embeddings for question-based retrieval, showing superior performance over classical methods. The study also introduces the LoCAL-bench dataset, comprising multimodal documents from nine domains, and demonstrates state-of-the-art results on public benchmarks, highlighting its efficiency and effectiveness for long-document understanding.

**Strength:**
- Proposes a novel framework to enhance the ability of LMMs for understanding long, visually-rich documents.
- Implements dual LoRA adapters for evidence page retrieval and question answering.
- Experiments are thorough, comparing the approach with both LLM-based and non-LLM-based baselines.

**Weakness:**
- The paper needs to compare LoCAL with additional methods (e.g., bge-large, NV-Embed-v2, SigLIP, ColPali) and evaluate its generalization ability across downstream tasks
- LoCAL-Bench lacks characteristics of long documents, limiting its ability to fully evaluate LoCAL's performance for multi-page understanding.
- It focuses on introducing and evaluating the LoCAL framework without a direct comparison to RAG methods.

**Additional Comments On Reviewer Discussion:**

Four reviewers highlighted several weaknesses in the work, particularly regarding generalizability and comparisons to RAG methods. The authors have addressed these concerns and revised the paper accordingly. While there is no strong champion for the work, all reviewers have provided positive feedback. Therefore, I recommend accepting this paper as a poster.

---

### Decision · Program_Chairs · 2025-01-22

Accept (Poster)

---

> ### Public Comment · ~Jian_Chen9 · 2025-03-01
>
> We appreciate the ethics review committee’s feedback. In response, we have added an Ethics Statement at the end of the main text, incorporating the discussion from our global response to Q2 in the rebuttal. We acknowledge and accept full responsibility for any legal implications related to our collected benchmark. Additionally, we have integrated the necessary clarifications from the rebuttal into the camera-ready version to ensure transparency and alignment with ethical guidelines. Furthermore, we have renamed our proposed method and benchmark dataset to better reflect their functionality and use case.